# Assessing the effectiveness of malaria interventions at the regional level in Ghana using a mathematical modelling application

Timothy Awine [1]*, Sheetal P. Silal [1,2]

**1** Modelling and Simulation Hub, Africa, Department of Statistical Sciences, University of Cape Town, Cape Town, South Africa, **2** Nuffield Department of Medicine, Centre for Global Health and Tropical Medicine, University of Oxford, Oxford, United Kingdom

* awinetimothy@gmail.com

## Abstract

Supporting malaria control with interfaced applications of mathematical models that enables investigating effectiveness of various interventions as well as their cost implications could be useful. Through their usage for planning, these applications may improve the prospects of attaining various set targets such as those of the National Strategic Plan policies for malaria control in Ghana. A malaria model was adapted and used for simulating the incidence of malaria in various regions of Ghana. The model and its application were developed by the Modelling and Simulation Hub Africa and calibrated using district level data in Ghana from 2012 to 2018. Average monthly rainfall at the zonal level was fitted to trigonometric functions for each ecological zone using least squares approach. These zonal functions were then used as forcing functions. Subsequently, various intervention packages were investigated to observe their impact on averting malaria incidence by 2030. Increased usage of bednets but not only coverage levels, predicted a significant proportion of cases of malaria averted in all regions. Whereas, improvements in the health system by way of health seeking, testing and treatment predicted a decline in incidence largely in all regions. With an increased coverage of SMC, to include higher age groups, a modest proportion of cases could be averted in populations of the Guinea savannah. Indoor residual spraying could also benefit populations of the Transitional forest and Coastal savannah as its impact is significant in averting incidence. Enhancing bednet usage to at least a doubling of the current usage levels and deployed in combination with various interventions across regions predicted significant reductions, in malaria incidence. Regions of the Transitional forest and Coastal savannah could also benefit from a drastic decline in incidence following a gradual introduction of indoor residual spraying on a sustained basis.

## Background

Countries with high malaria burden have over the years fashioned out strategies with the goal to eliminate the disease. However, most of these countries will need to establish control of

**Data Availability Statement:** All data,codes and supporting files are made available at https://zenodo.org/badge/latestdoi/531038462.

**Funding:** This research was funded in whole, or in part, by the Wellcome Trust [Grant number 2114236/Z/18Z]. For the purpose of open access, the author has applied a CC BY public copyright licence to any Author Accepted Manuscript version arising from this submission. The research also forms part of TA's PhD studies sponsored by SACEMA and the Ghana Education Trust Fund (GetFund). The funders had no role in study design, data collection and analysis, decision to publish, or preparation of the manuscript.

**Competing interests:** The authors have declared that no competing interests exist.

malaria first before contemplating the achievement of elimination within the shortest time frame with limited resources at their disposal. This often leads to a concurrent deployment of multiple interventions with the aim of achieving set targets as exemplified in most sub-Saharan Africa countries where the disease is endemic.

In Ghana, the national strategic policy for malaria control (NSP) outlines a number of strategies to be deployed within the 2014–2020 period with the goal of achieving a 75.0% reduction in malaria morbidity and mortality. Among the objectives are the deployment of malaria preventative measures to cover at least 80.0% of the population, testing all (100%) suspected cases of malaria with the aim to subsequently treat all confirmed cases promptly and rolling out Seasonal Malaria Chemoprevention (SMC) in targeted districts of the Guinea savannah zone [1, 2].The success of achieving these targets will largely depend on how well interventions are rolled out at the regional level, enhanced by knowing their effectiveness.

The Ghana National Malaria Control Program (NMCP) however previously stated that, it lacks the bases to justify the deployment of these interventions across Ghana and also lacks the evidence to support their impact. Additionally, the NMCP expressed concerns on the sustainability of successes chalked so far and possible funding support to implement its programs going forward in the face of a declining counterpart funding from international agencies [3].

As stated elsewhere [3], some of these challenges could potentially be addressed with the support of mathematical modelling tools, which date back several years. The usefulness of mathematical models encompasses many fields including public health. For instance they can be developed to assist with disease control policy formulation, monitoring and evaluation of disease incidence, elucidate on transmission dynamics of diseases and used to assess the cost effectiveness of optimising the impact of interventions, deployed singly or in combination in regions of varying transmission dynamics such as Ghana [4]. Since interventions are mostly deployed along the regional administrative areas in Ghana, which in turn are located in the various transmission zones, it is imperative for these investigations to be conducted on regional basis in order for the findings to be more informative.

In this study, a model application developed by the Modelling and Simulation Hub, Africa (MASHA) group, was used to simulate the impact of malaria interventions regionally in Ghana. The application allows for various intervention scenarios as well as their associated costs to be investigated interactively for different time periods. It also allows for easy analyses of the cost effectiveness of the various interventions under various scenarios for different regions [5, 6].

However, to successfully use the application to design an accelerated regional strategy for malaria elimination in Ghana, the basic seasonality module was modified to account for the different transmission settings in Ghana. Thus, three seasonality functions were developed and applied to all regions in the three respective transmission settings across Ghana.

## Methods

The underlying mathematical model is a single patch model for simulating *Plasmodium falciparum* malaria transmission. The model is a compartmental model involving four classes of infection types namely; severe, clinical, asymptomatic detectable and undetectable by microscopy. Estimation of the sensitivity of various tests is carried out using the associated levels of parasitaemia with the different infection classes [7].

### Ethics

Aggregated out-patients data from health facilities together with monthly average rainfall data were used for fitting the models on the application. This study obtained ethical clearance from

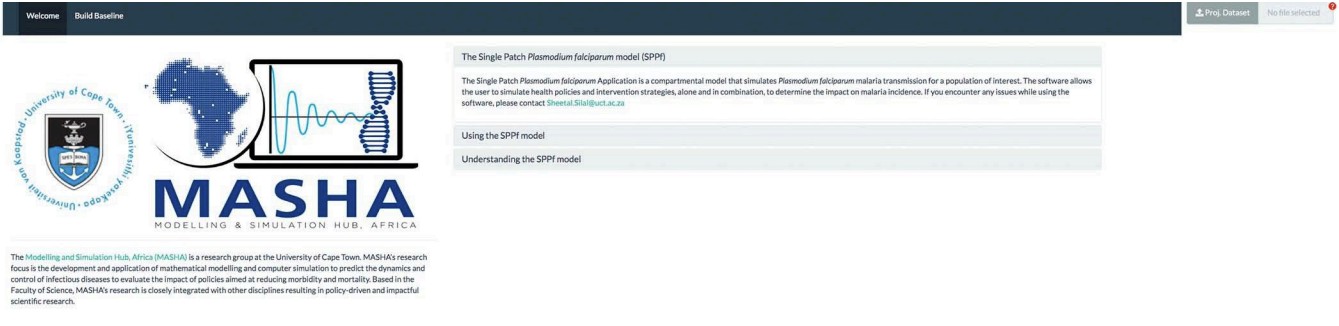

**Fig 1. Dashboard of the single patch *Plasmodium falciparum* model interface in Rshiny.**

the Navrongo Health Research Centre IRB, The ethics committee of the Science Faculty of the University of Cape Town and permit to use the data from the National Malaria Control Program of the Ghana Health Service. The data being used does not include individual identities.

## Application dashboard

The application is presented in a dashboard format and supporting documents and the code hosted at [8]. As shown in Fig 1, prospective users of the software application have the opportunity to simulate health policies and intervention strategies, alone and in combination, to determine their impact on malaria incidence, for a given region or province of a country.

On the home page, there are two menus; the "Welcome page", which provides information about the application and the "Baseline page" that allows for the inputs of baseline data and parameters for subsequent calibration.

## Running the application

A number of steps are involved in running the application in R. The first step being setting up the baseline epidemiological values for a population of interest, the second to validate the outputs with observed data, thirdly conduct investigations on the impact of various interventions and subsequently download predicted outputs and associated data for each intervention tested.

The costs of most of the interventions have already been studied in Ghana at the zonal level using this application. The costs associated with the interventions investigated in this study have therefore not been considered [7].

**Baseline settings.** Once the application is compiled in R shiny the "Build Baseline" menu appears on the top left corner of the dashboard. Among the many parameters to be imputed under this menu are population size and growth rate, Annual Parasite Incidence (API) per 1000 population (approximated using the incidence/1000 population) per year and baseline coverage levels of various interventions as outlined in S2 Table.

After calibrating the baseline data inputs, a menu for validating the results will be displayed. Additional menus will display for various intervention scenarios to be simulated once the validation process is completed. These menus include, Simulate Interventions, Explore and Sensitivity Analysis, as shown in Fig 2.

**Simulating interventions singly or in combination.** After the necessary input data have been calibrated, the impact of various intervention packages can be simulated for different regions of populations. For instance on a singly bases, the impact of improved coverage or usage levels of Insecticide Treated Bednets (ITN) could be investigated. Likewise the other interventions such Indoor Residual Spraying (IRS). The Application currently is capable of

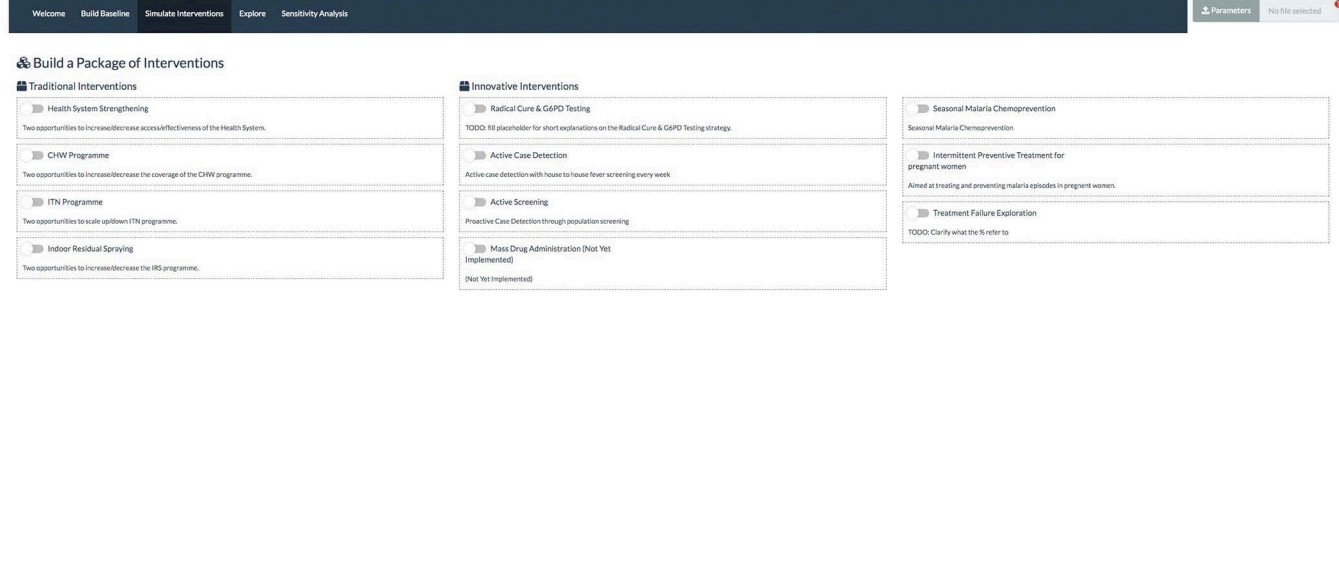

**Fig 2. Interface for setting values of each intervention scenario to be tested.**

being used to investigate the impact of strengthening the health system (HSS) and well as increasing the coverage levels of Community Based Health Planning and Services (CHPS) or Community Health Workers (CHW). Investigations into the impact of SMC in regions suitable for its deployment are also possible.

The impact of these interventions could also be investigated in combination. For example enhanced ITN usage could be combined with a more effective insecticide used for IRS or expanded SMC or together with an improved health system with an increase in the probability of being tested and once confirmed to have malaria treated promptly.

Depending on the intervention to be investigated, the various menus allows for two opportunities of switching to scale up, for various periods in years of coverage levels and levels of enhanced efficacy where applicable.

## Contribution to the SPPf app

The need to modify the seasonal functions and other features of the application to make them suitable for modelling dynamics of malaria morbidity in the various regions are described in the following sections.

**Current seasonal forcing function of the application.** The seasonal forcing function included in the app is a cosine function that generates a single peak. The function as shown by Eq (1) is unable to replicate the transmission patterns in areas where the incidence of malaria peaks twice a year such on the Transitional forest and Coastal savannah zones.

$$(1 + T2 * \cos(2 * \pi * (t - \emptyset)))^{pe} \tag{1}$$

Where **T2** is the amplitude which is allowed to vary, **t** model step time, **phi** the phase angle and **pe** representing the "peakness" of incidence cases. Additionally, the current function is

unable to capture the "peakness" of malaria incidence exhibited in some of the regions of the Guinea savannah zones.

These shortfalls therefore necessitated some modifications be made to the seasonality function in the application to allow for zonal specific seasonal forcing functions to be applied to regions in these zones.

## Modifications to the seasonal function of the application

As shown on Fig 3 panel (a), (b) and (c) respectively for regions in the Guinea savannah, Transitional forest and Coastal savannah, the patterns of malaria incidence (grey bars) are seasonal and generally lag behind the rainfall incidence (blue line).

Seasonality also varies across the ecological as the rainfall peaks only once in the Guinea savannah but twice a year in the Transitional forest and Coastal savannah [2].

The dynamics of the seasonal forcing function used in the application is shown on Fig 4 panel (a). The inadequacy of the simple cosine function in modelling the diverse transmission dynamics across Ghana is shown in the single peak the function exhibits. To successfully apply this modelling application at the regional level in Ghana therefore requires modifying the seasonal forcing function in Eq (1) implemented by the application to functions that reflect the patterns of morbidity of malaria in the various regions of the various ecological zones. The seasonal forcing functions created were therefore modified cosine functions that were fitted to monthly rainfall data for each zone.

The regions within the savannah transmission zone include, per the old regional demarcations were, Upper East, Upper West and Northern. Similarly those within the Transitional forest zone were Ashanti, Bono-Ahafo, Eastern and Volta. Greater Accra, Central, and Western regions belong to the Coastal transmission zone.

**Seasonal forcing functions developed by zone.**   The seasonal forcing functions were modified with standardised average monthly rainfall data from three different transmission zones of Ghana. These functions as in Eqs (2), (3) and (4) for the Guinea savannah, Transitional forest and Coastal savannah zones respectively to together with respective administrative regional clinical data, collected through the DHIMS platform were used to calibrate the models for each of the 10 regions (old regional administrative boundaries) of Ghana.

The seasonal forcing functions were obtained through least squares fitting approach using the Optim package in R [9].

$$\mathbf{S} = (\mathbf{T1} + \mathbf{T2} * \cos(2 * \pi * \mathbf{t} - \emptyset))^{pe} \tag{2}$$

$$\mathbf{F} = (\mathbf{T1} + \mathbf{T2} * \cos(2 * \pi * \mathbf{t} - \emptyset_1) + \mathbf{T3} + \mathbf{T4} * \cos(4 * \pi * \mathbf{t} - \emptyset_2))^{pe} \tag{3}$$

$$\mathbf{C} = (\mathbf{T1} + \mathbf{T2} * \cos(2 * \pi * \mathbf{t} - \emptyset_1) + \mathbf{T3} + \mathbf{T4} * \cos(4 * \pi * \mathbf{t} - \emptyset_2))^{pe} \tag{4}$$

Where **S** is the forcing function for regions of Guinea savannah zone, **F** for the Transitional forest zone and **C** the Coastal Savannah. **T1** and **T3** are constants and **T2** and **T4** the amplitudes, both in the range (0, 1) and **phi** is the phase shift parameter and **t** times of the simulation time step with **pi**, a constant. **pe** is the power to which the function must be raised in order to achieve the desired peak compared to the observed data.

The respective fitted monthly rainfall data and the observed data for each zone yielding optimal parameters of the functions are shown on below in Fig 4. Function parameter values are shown on Table A in the S1 Table.

The functions generated allowed the models to be fitted individually for each region for all three zones. Thus, the incidence of malaria in regions of the Guinea savannah zone peak once

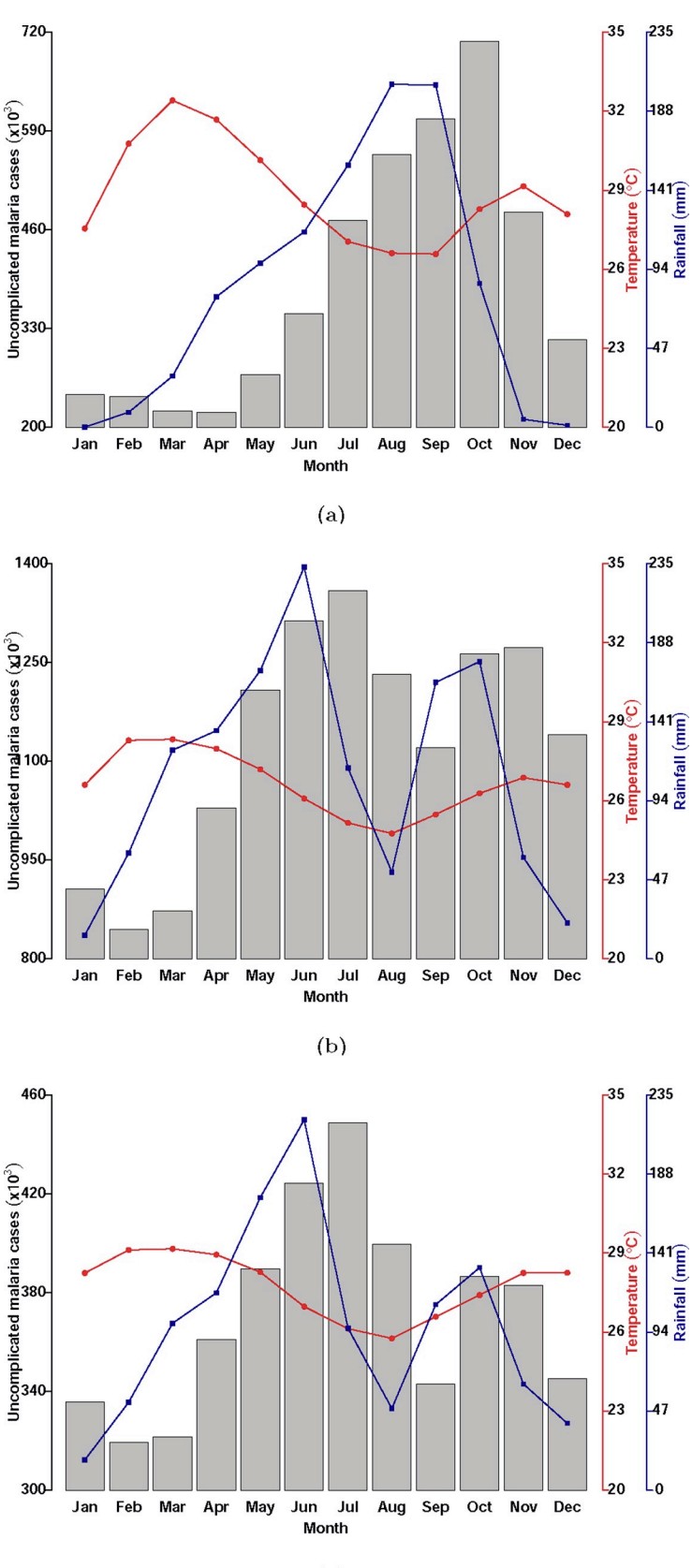

**Fig 3.** Patterns of malaria incidence by month of year (grey bars), monthly average rainfall (blue line) and average monthly temperature (red line) by ecological zone [(a) Guinea savannah, (b) Transitional forest and (c) Coastal savannah].

as the does rainfall (Fig 4 panels (b)), and those of the Transitional forest and Coastal savannah experience two peaks, the first being more pronounced than the second (Fig 4 panels (c) and (d)).

*Seasonal forcing functions input sliders.* As part of the changes, additional sliders were created to accommodate the input parameters for the functions. For the Guinea savannah model, two sliders (**amplitude1** and **amplitude2**) were provided for the amplitude parameters.

As shown on Table 1, the cosine function obtained for the regions of the Guinea savannah allows for adjustments to be made to two amplitude parameters as in **amplitude1** and **amplitude2** as well as changes to the phase angle **phi**. The "peakedness" of the function that allows for a possible replication of the observed clinical data for a given region is adjusted through **pe**, Fig 5.

Similarly, seasonal forcing functions for regions in the Transitional and Coastal savannah, are made of two cosine functions, the first corresponding to the major season and the second for the minor transmission seasons, respectively.

Adjustments, one each for the two combined functions, are allowed to be made on two amplitudes through **amplitude1** and **amplitude2** as well as for the two phase angles as in **phi1** and **phi2** respectively as shown on Fig 6.

Whiles one input slider (**peak month1**) was included in the function for the Guinea savannah to allow for the values of the phase angle, two sliders (**peak month1** and **peak month2**) were added for inputting two phase angles for seasonal functions of the Transitional forest and Coastal savannah zones respectively.

## Using the application at the regional level in Ghana

The interventions that were simulated in the various regions of Ghana were health system strengthen (HSS), improving coverage levels of Community Health Workers/Community–Based Health Planning and Services (CHW/CHPS), improving on the coverage and effectiveness of ITNs and Indoor Residual Spraying (IRS) in all regions of the country and rolling out SMC in regions of the Guinea savannah. The scenarios investigated by region were:

**Health system strengthening.**

- Increasing the probability if health seeking to 90%

- Increasing the probability if being tested at the health facilities to 95%

- Increasing the of being treated if tested to be positive to 90%

**Strengthening the CHW/CHPS by.**

- Increasing the coverage of CHW/CHPS to 90%

- Increasing the probability if health seeking to 90%

- Increasing the probability if being tested at the health facilities to 95%

- Increasing the of being treated if tested to be positive to 90%

**Distribution of ITN.**

- Increase the coverage of ITNs to 90%

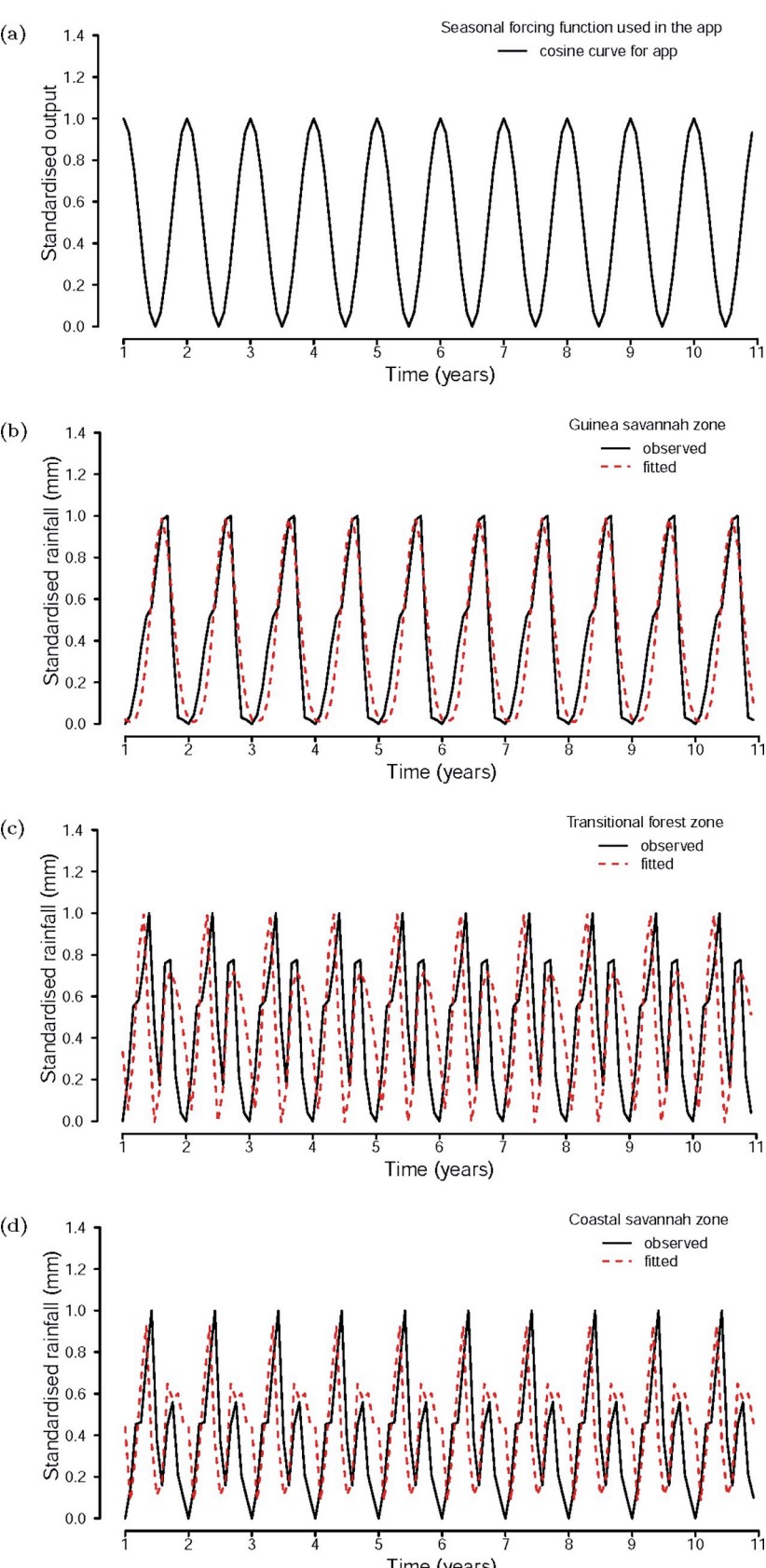

**Fig 4.** Panels (a), (b), (c) and (d) represent the application cosine output curve, fitted rainfall data for the Guinea savannah, Transitional forest and Coastal savannah zones respectively.

- Improve on the effectiveness of the ITNs through the improvement on usage to 50%

  **IRS implementation.**

- Improve on sustained coverage to 90%

- Avoidance of biting due to IRS by adopting practices that lead to a 40% reduction in biting rate

  **SMC in regions of the Guinea savannah.**

- Increase coverage to 20% to include children from 3 month to 10 years.

These scenarios were selected to include an increment of 10% - 20% points, based on their respective levels at baseline year (2018). The interface that allows for inputting values for the various scenarios is shown on Fig 2.

Additionally, the interventions were tested in combination with enhanced ITN usage of 75% while maintaining coverage at 90%, except for the Upper West region where ITN usage of 90% was tested.

In combination with IRS, efficacy was maintained at 30% (increased to 50% for Upper West region) while those of HSS, CHW/CHPS and SMC maintained as in the scenarios tested above. The combined approach was to investigate which combination allows for the attainment of the milestones of the NSP 2014–2020 especially or those of WHO on pre-elimination conditions.

## Model parameters and source

Model parameters for malaria incidence with respect to each region were mostly sourced from reports of the NMCP as shown on parameter tables in S2 Table.

## Results

### Guinea savannah zone

Baseline incidence rates in 2018, for the three regions of the Guinea savannah zone, were estimated to be 373, 321, and 146 per 1000 population respectively in the Upper East, Upper West and Northern regions.

On the basis of deploying interventions singularly and after adjusting for baseline (2018) coverage levels of the various interventions, the proportion of reported incidence cases of malaria by 2030 in the Upper East region was predicted to be 0.9% for CHW/CHPS only and an increase of 21.7% of incidence cases seen for HSS improvement. The proportion of cases

**Table 1. Code for seasonality function applied to the three transmission settings across Ghana.**

| Zone | Code |
|------|------|
| Guinea savannah | $seas <- (amplitude1 + amplitude2*cos(2*pi*t\_internal—(phi + 8)))\^pe$ |
| Transitional forest | $seas <- (-11.05929764 + amplitude1 *cos(2*pi*t\_internal—phi1) + 10.89390065 + amplitude2 *cos(4*pi*t\_internal—phi2))\^pe$ |
| Coastal savannah | $seas <- (48.54206007 + amplitude1 *cos(2*pi*t\_internal + phi1) + -48.71330976 + amplitude2 *cos(4*pi*t\_internal—phi2))\^pe$ |

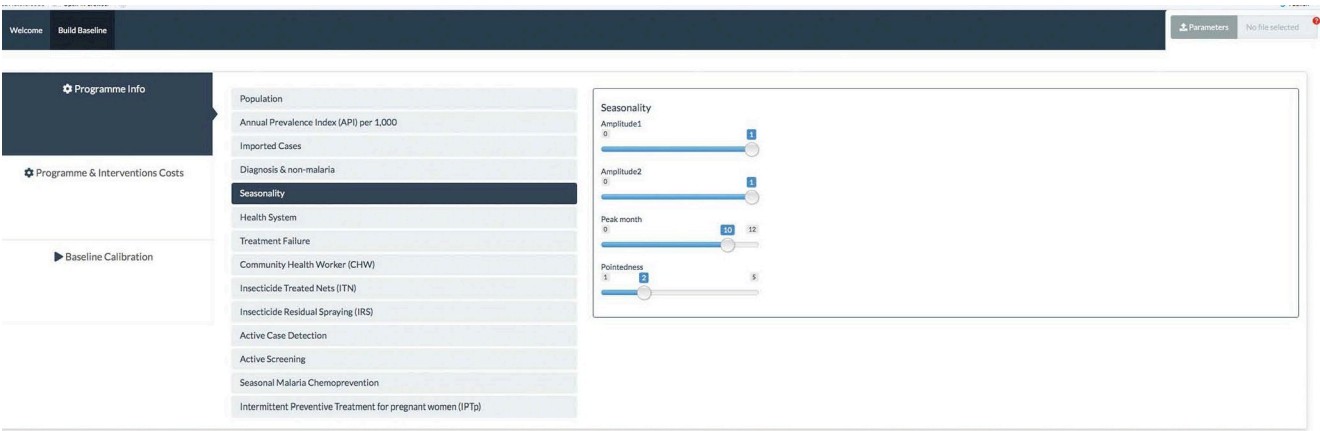

**Fig 5. Changes to input sliders for amplitude parameters of the Guinea savannah model code.**

averted following the deployment of ITN was 9.3%, 5.5% for IRS and 5.6% for SMC among children under 10 years, Table B in S1 Table and Fig 7A1.

With respect to malaria incidence per 1000 population by 2030, 337 for ITN, 350 for IRS and 353 for SMC respectively were predicted for single intervention deployment in the Upper East region, Table B in S1 Table.

Similarly, for the Upper West region, the proportions of reported cases seen at the health facilities following an improved HSS and an increased coverage of CHW/CHPS respectively were 41.9% and 0.0%. Reported cases averted by deployment of ITN alone was 7.8% and 0.4% for IRS alone whiles for SMC alone it was 5.4%. The corresponding incidence rates/1000 population were 294, 319 and 304 respectively for ITN, IRS and SMC on single deployment basis, Table B in S1 Table and Fig 7B1.

The proportion of cases averted by 2030 after implementing the various packages of interventions on a single basis in the Northern region were 79.3%, 51.3% and 40.8%, respectively for ITN, IRS and SMC. In this region, an improved health system predicts 61.7% cases averted by 2030 and 0.1% with an increased coverage of CHW/CHPS. These reductions in terms of incidence were 13/1000, 50/1000 and 80/1000 for ITN, IRS and SMC respectively as shown on Table B in S1 Table and Fig 7C1.

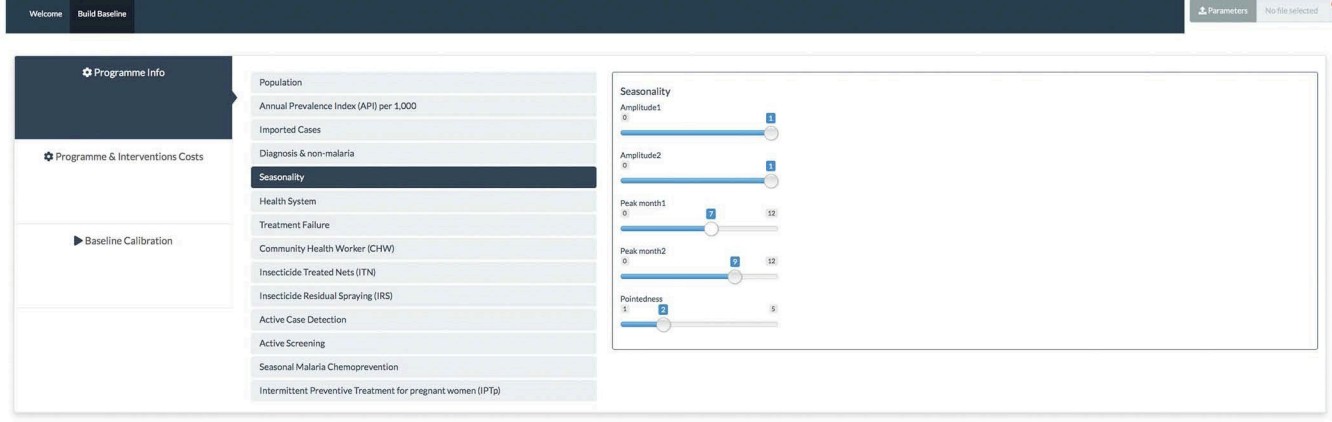

**Fig 6. Changes to input sliders for amplitude parameters of the Transitional forest and Coastal savannah model codes.**

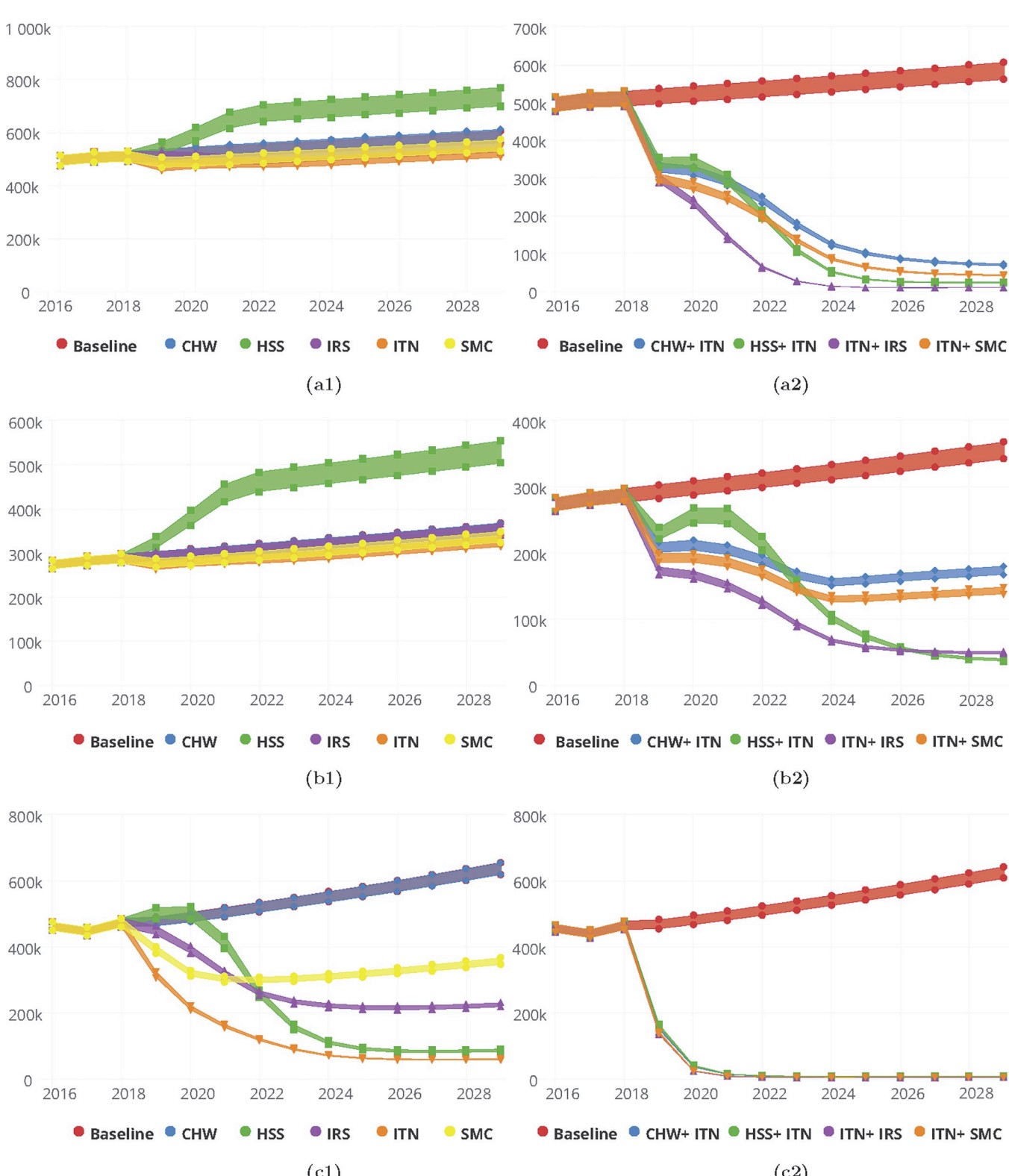

**Fig 7.** Predictions for malaria incidence for various intervention packages, singly (a1-c1) for Community Health Worker (CHW), Health System Strengthening (HSS), Indoor Residual Spraying (IRS), Insecticide Treated bednets (ITN) and Seasonal Malaria Chemoprevention (SMC) and in combination (a2-c2) for CHW/CHPS + ITN, HSS + ITN, ITN + IRS and ITN + SMC for the Upper East (a1 and a2) Upper West (b1 and b2) and Northern (c1 and c2) Regions respectively.

Predicted proportions of cases averted by 2030 for a combined deployment of the interventions with an enhanced ITN usage at 90% and coverage at 90% were predicted to be 68.9%, 76.0%, 86.4% and 75.4% in the Upper East region respectively for CHW/CHPS + ITN, HSS +ITN, IRS +ITN and SMC +ITN. Similarly in the Upper West region they were 44.6%, 59.3%, 71.0% and 52.2% and 95.8%, 95.4% 96.6% and 96.4% for the Northern region respectively for CHW/CHPS + ITN, HSS +ITN, IRS +ITN and SMC +ITN, Table E in S1 Table and Fig 7A2–7C2.

## Transitional forest zone

The baseline incidence (2018) for the regions in this zone were estimated to be 140, 285, 211 and 170 per 1000 population respectively for the Ashanti, Brong-Ahafo, Eastern and Volta regions.

Predicted proportions of incidence cases averted in the Ashanti region for the various interventions deployed singly after simulation were estimated to be 67.9%, 48.2% and 53.6% respectively for ITN, IRS, HSS. The impact of CHW/CHPS was however 0.1%. The corresponding incidence/1000 population predicted by 2030 for ITN, IRS and HSS were observed to be 14, 57 and 37 respectively, Fig 8A1 and Table C in S1 Table.

In the Brong-Ahafo region, Fig 8B1 and Table C in S1 Table, the cases averted by 2030 were predicted to be 18.3%, and 10.5% respectively for ITN and IRS when deployed on a singularly. The improvement in the health system (HSS) predicted an increase in the number of incidence cases reported in the health facilities by 36.9% and 0.2% for an increase in the coverage of CHW/CHPS.

While HSS predicted 36.8% reduction in the reported cases of malaria by 2030 in the Eastern region, CHW/CHPS predicted a 0.1% increase in the number of reported incidence cases captured. The proportion of cases averted in the Eastern region following deployment of ITN and IRS were 67.5% and 37.1% respectively. Correspondingly the incidence rates per 1000 population predicted were 43 and 116 respectively for ITN and IRS, Fig 8C1 and Table C in S1 Table.

The predicted cases averted in the Volta region by 2030 were 72.5%, 52.2%, 52.6% and 0.1% for ITN, IRS, HSS and CHW/CHPS respectively. These reductions were associated with incidence rates of 34, 59, 47 and 170 per 1000 population respectively by 2030, Fig 8D1 and Table C in S1 Table.

The combined deployment of the interventions with enhanced ITN usage as applied in the regions of the Guinea savannah predicted not less than 80.0% cases averted for all three combinations of the interventions. Incidence rates per 1000 population corresponding to these reductions ranged from 20 to 31, Fig 8A2–8D2 and Table F in S1 Table.

## Coastal savannah zone

At baseline (2018) incidence per 1000 population were 220, 47 and 249 for the Central Greater Accra and Western regions respectively.

Applying the same intervention scenarios for single intervention deployment as with the rest of the regions, leads to a predicted proportion of 47.4%, 85.4% and 40.6% of cases averted by ITN respectively in the Central, Greater Accra and Western regions. With respect to IRS, the predicted proportions were 26.0% 69.1% and 19.6% respectively. While HSS in the Greater Accra leads to a predicted 64.0% reduction in the cases of malaria seen, 4.1% and 24.3% more cases were predicted to be captured at the health facilities in the Central and Western regions respectively by 2030. Reductions in incidence rates less than 10/1000 population were predicted for ITN, IRS, HSS in the Greater Accra Fig 9A1–9C1 and Table D in S1 Table.

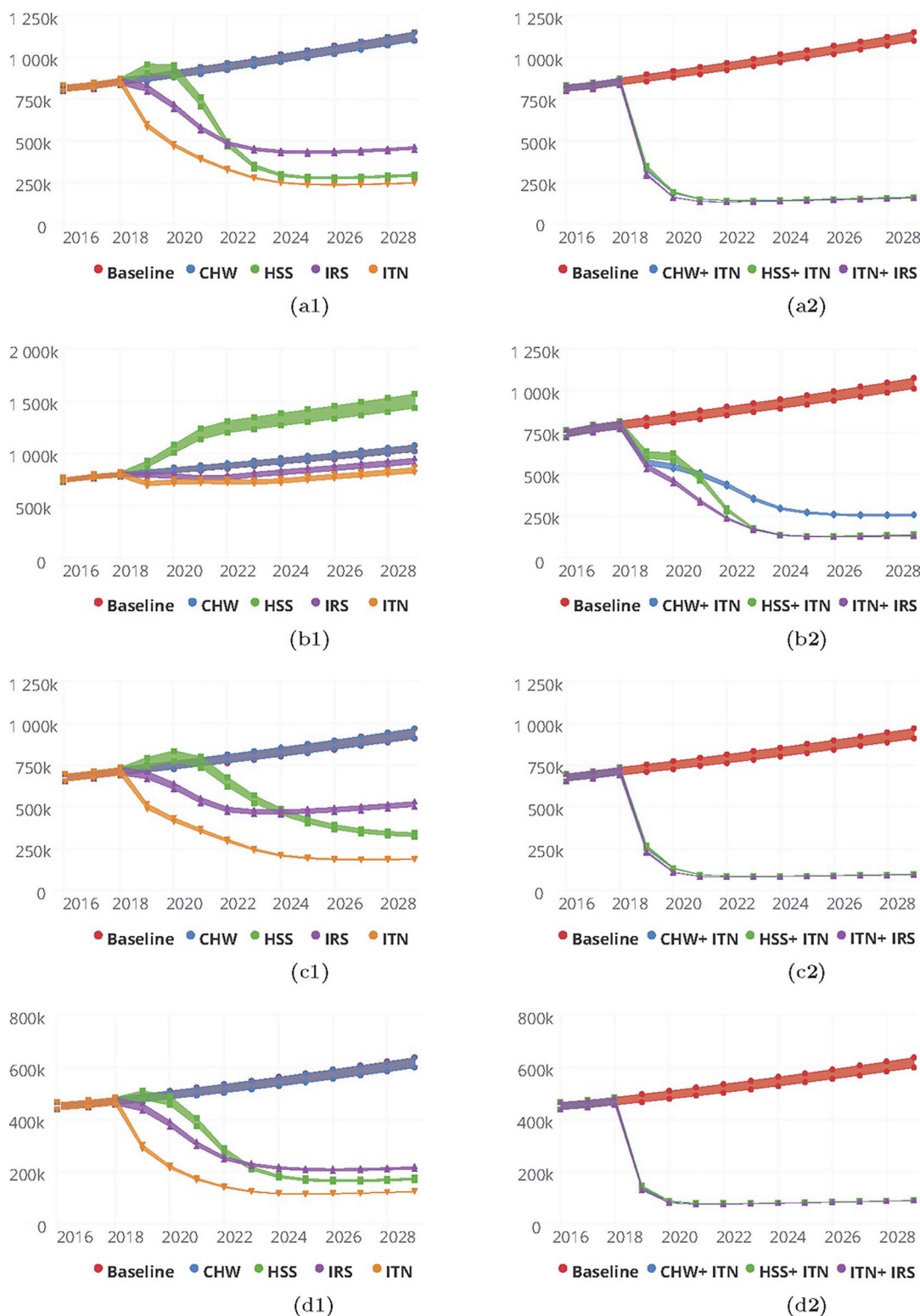

**Fig 8.** Predictions for malaria incidence for various intervention packages, singly (a1-d1) for Community Health Worker (CHW), Health System Strengthening (HSS), Indoor Residual Spraying (IRS), and Insecticide Treated bednets (ITN) and in combination (a2-d2) for CHW/CHPS + ITN, HSS + ITN and ITN + IRS for the Ashanti (a1 and a2), Brong-Ahafo (b1 and b2), Eastern (c1 and c2) and Volta (d1 and d2) Regions respectively.

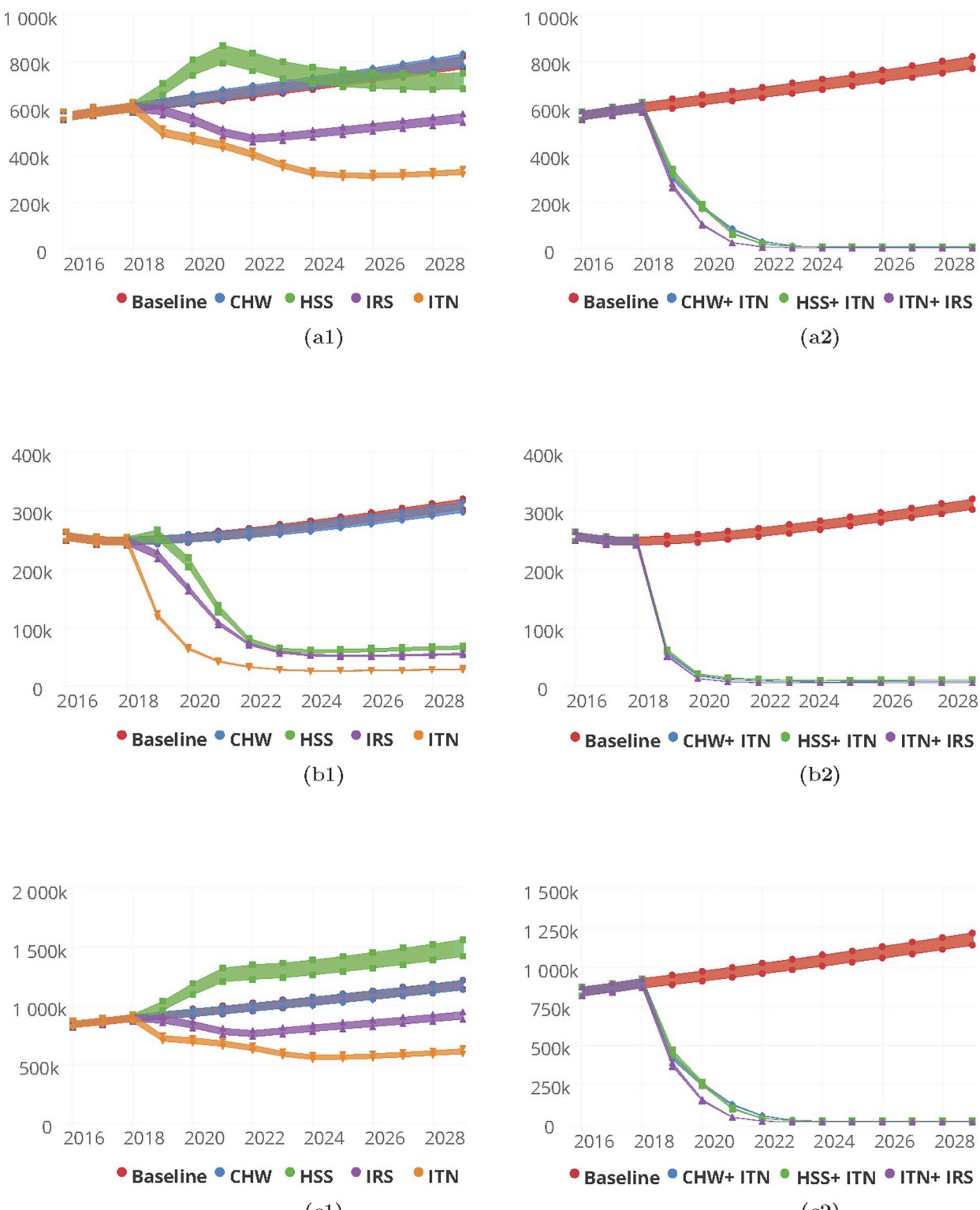

**Fig 9.** Predictions for malaria incidence for various intervention packages, singly (a1-c1) for Community Health Worker (CHW), Health System Strengthening (HSS), Indoor Residual Spraying (IRS), and Insecticide Treated bednets (ITN) and in combination (a2-c2) for CHW/CHPS + ITN, HSS + ITN and ITN + IRS for the Central (a1 and a2), Greater Accra (b1 and b2) and Western (c1 and c2) Regions respectively.

Predictions for similar combined interventions implemented with enhanced ITN usage as in the other regions showed that, incidence rates in the range 1 to 3 per 1000 population were predictable by 2030 for all the regions in this zone. Corresponding proportion of cases averted predicted were similarly above 90.0% across all regions in this zone for CHW/CHPS + ITN and IRS + ITN, and HSS +ITN Fig 9A2–9C2 and Table G in S1 Table.

## Discussions

A malaria modelling application, developed for *P. falciparum* malaria, has been used to simulate various interventions scenarios at the regional level in Ghana for the first time.

The model was calibrated and validated using monthly data from different transmission settings in Ghana. Data each transmission zone from 2012 to 2017 years were used to calibrate the model. Each zonal model was then validated using data from 2017 for a year. As shown in the description file in [8], the seasonal patterns of malaria incidence in the various zones have been captured and 50.0% projected uncertainty range captures most of the observed data as well within the validation range.

The ability to use it to speedily evaluate the impact of various interventions and their accompanying costs at both the national and sub-national levels without requiring any additional resources makes them potentially useful in supporting the policy decision-making process with respect to malaria control in any population of interest.

Previous studies have reported the costs associated with a number of malaria interventions in Ghana but these were mostly subsequent to expensive field trials and mainly focused on one or two interventions [10, 11]. However, this application was recently used to study the costs associated with malaria interventions in Ghana at the zonal level [7].

In this study various interventions were considered and their impact on the incidence of malaria were observed to vary across the regions largely due to prevailing baseline coverage levels and level of local transmission potential (incidence rates/1000 population).

For instance in regions such as Upper East, Upper West and the Brong-Ahafo, where the API is greater than 250, strengthening the health system through improvement of health seeking, diagnosis and treating of suspected cases (HSS) predicts an increase in the annual number of cases reported and treated compared to the other regions such as the Northern, Ashanti and Greater Accra, which reported relatively lower API. In these regions, of relatively lower API, improvement in HSS rather predicted a rapid decline in incidence of malaria, in some cases after an initial increase as shown on Figs 7C1, 8A1, 8C1, 8D1 and 9A1–9C1.

Further simulations investigating the impact of combining various interventions in regions of relatively higher API seems to suggest deployment of the interventions in combination averts more cases of malaria. For example further declines in incidence of malaria were predicted with the deployment of HSS, IRS, SMC and CHW/CHPS in combination with enhanced ITN usage (90.0%).

For example, in a scenario where HSS was deployed in combination with enhanced ITN usage predicted, 76.0%, 59.3% and 80.9% of cases are averted over a 12-year horizon compared to either intervention deployed alone in the Upper East, Upper West and Brong-Ahafo regions respectively. Similarly deployment of IRS in combination with enhanced ITN usage predicted 86.4% and 71.0% of cases averted whereas SMC with enhanced ITN usage deployed in combination predicted 75.4% and 52.2% reductions in incidence cases respectively for Upper East, Upper West regions.

Investigations on the impact of ITN suggest that, improved usage impacts much more on averting incidence of malaria other than just an increase in the coverage. The usage level investigated was at least a doubling of the baseline usage levels in each region. Following earlier

results at the zonal level, this seems to be a consistent finding in the various regions [12]. Emphasising on the usage of ITNs while coverage is sustained should therefore be prioritised.

As with all other interventions deployed singly, IRS deployed alone predicts higher reductions in incidence over time in almost all regions. However, the reductions predicted may not be enough to reach pre-elimination status in any of the regions. Achieving program goals and targets such as those of the NSP will require a combination of the interventions with an enhanced coverage and usage for ITN, which is the most accessible of the list of possible interventions covered in this study [1].

Pre-elimination milestones seem to be achievable in the Northern region and as well as all regions of the Coastal savannah for all the combined interventions, Tables E and G in S1 Table, Figs 7C2 and 9A2–9C2.

Across all the regions, a mere increase in the coverage levels of CHW/CHPS, does not impact highly on the incidence of malaria. It does seem, an increase in coverage with an accompanying health systems improvement by way of increased probability of seeking health and getting tested and treated once malaria is confirmed for all suspected cases leads to a more appreciable proportion of cases averted.

Presently IRS is largely rolled out in selected districts of the Guinea savannah (Upper East, Upper West and Northern regions). The results for the regions in the Transitional forest and Coastal savannah shows that, these regions could also benefit from the deployment of IRS. This is evidenced from the proportion of cases averted following the deployment of IRS at 90% coverage in the Ashanti, Brong-Ahafo, Eastern, Volta, Central Greater Accra and Western regions.

The predictions made into the future to 2030 were based on the assumptions that, current environment and vector behaviour remains constant going into the future. With respect to vector behaviour, recent studies have suggested an emerging threat of insecticide resistance particularly for pyrethroids in some parts of Ghana. Unfortunately the use of this insecticide for both pests and vector control form part of the main strategies of vector control activities through treated bed nets and IRS. The impact of this development may be minimised or averted if alternate insecticides are deployed for both treated bed nets and IRS [13, 14]. This notwithstanding, the use of treated bednets and IRS may still proffer some protection to the larger rural population once they are used effectively [12, 15, 16].

Having modified the seasonal function of the application to reflect zonal malaria morbidity dynamics allows the application to be used to predict more accurately, the patterns and impact of various malaria intervention packages across different transmission settings in Ghana. However, assuming a constant rainfall and temperature is a limitation that could render the model to produce inaccurate results in the prediction window should the assumption fail. For instance if any of the zones experiences a severe draught or unusually high volumes of rainfall in the future as a result of climate change effects, this could impact on the predicted number of cases. Nevertheless, the forcing functions could be updated using locally reported weather data. Another limitation that arises is in not accounting for varying household socio-economic levels across zones in the predictions made. It has been shown that, increase risk of malaria infection is highly associated with the wealth of the household [17]. The effect of this association on the impact of these interventions may however be averaged out at the population level.

The relative speed of use and easy visualisation of model outputs afforded by the application should make it easy for disease control managers to use to guide policy planning and intervention deployment. The single patch nature of the application does not allow it to incorporate cross regional interactions of the population that may account for any importation of malaria infected cases. However, the lack of this feature in the application may have a low impact given

the relatively high prevalence of malaria across all regions. Additionally, due to the lack of age structure, investigations of the burden of malaria among various age groups is not possible.

Key features of malaria epidemiology in an endemic country such as Ghana have been taken account of in the model. These features include superinfection and immunity. Given the high levels of malaria incidence in all three zones, the model outputs may be greatly improved having these key features incorporated even though the full knowledge of how immunity works is not yet understood [18]. Additionally the model accounts for the various stages of malaria infection and makes room for diagnostic sensitivity for these classes of infection. It incorporates adjustments for a possible El Nino event and provides the opportunity to be adapted for various transmission settings through the incorporation of monthly rainfall for a given location of interest.

The cost effectiveness of most the interventions investigated in this study have been considered elsewhere [7]. Even though the cost effectiveness of the interventions at the regional level could be important, the focus of this study was limited to investigating the impact of the various interventions and the results of the referred study may still be applicable at this level.

## Conclusions

Deployment of each intervention singly seems not be enough for any region to achieve pre-elimination targets and those of the NSP of 2014–2020. If the targets set by the NSP will be met in the various regions, intensifying the usage of ITNs while maintaining their coverage may boost in changing the dynamics of malaria incidence in the various regions. Additionally, sustaining the implementation of IRS campaigns in some selected districts in all the regions will be significant.

While increasing the coverage of CHW/CHPS will may lead to an improved health system, by increasing access in the communities, improvement upon health seeking, diagnosis testing and treatment of all confirmed cases will greatly strengthen the health system to promptly capture cases that were not prevented by vector control interventions that have been implemented due to poor usage or uptake at the personal level.

This modelling application if adopted by managers of malaria control in Ghana could offer invaluable support in policy design for planning future malaria control strategies at both the national and sub-national levels across different transmission settings.

## Supporting information

**S1 Table.**
(DOCX)

**S2 Table. Table of parameters and their source by region for model application.**
(DOCX)

## Acknowledgments

This work is based on the research supported by the Wellcome Trust, Department of Science and Innovation and the National Research Foundation. Any opinion, finding, and conclusion or recommendation expressed in this material is that of the authors and the NRF does not accept any liability in this regard."

## Author Contributions

**Conceptualization:** Timothy Awine, Sheetal P. Silal.

**Data curation:** Timothy Awine.

**Formal analysis:** Timothy Awine.

**Investigation:** Timothy Awine.

**Methodology:** Timothy Awine.

**Software:** Sheetal P. Silal.

**Supervision:** Sheetal P. Silal.

**Writing – original draft:** Timothy Awine.

**Writing – review & editing:** Sheetal P. Silal.

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
