## [Decision Letter · Decision Letter 0]

28 Jun 2022

PGPH-D-22-00584

Assessing the effectiveness of malaria interventions at the regional level in Ghana using a mathematical modelling Application

Dear Dr. Awine,

Thank you for submitting your manuscript to PLOS Global Public Health. After careful consideration, we feel that it has merit but does not fully meet PLOS Global Public Health’s publication criteria as it currently stands. Therefore, we invite you to submit a revised version of the manuscript that addresses the points raised during the review process.

Please submit your revised manuscript by . If you will need more time than this to complete your revisions, please reply to this message or contact the journal office at globalpubhealth@plos.org. Please include the following items when submitting your revised manuscript:

We look forward to receiving your revised manuscript.

Kind regards,

Muhammad Imran Nisar, MBBS, MSc

Academic Editor

Journal Requirements:

2. Please update your online Competing Interests statement. If you have no competing interests to declare, please state: “The authors have declared that no competing interests exist.”

3. We ask that a manuscript source file is provided at Revision. Please upload your manuscript file as a .doc, .docx, or .rtf.

4. Please ensure that you refer to Fig 6 in your text as, if accepted, production will need this reference to link the reader to the figure.

5. Please include a copy of Table 2 which you refer to in your text on page 6.

6. We have noticed that you have uploaded Supporting Information files, but you have not included a list of legends. Please add a full list of legends for your Supporting Information files after the references list.

Additional Editor Comments (if provided):

Reviewers' comments:

Reviewer's Responses to Questions

**Comments to the Author**

1. Does this manuscript meet PLOS Global Public Health’s publication criteria? Is the manuscript technically sound, and do the data support the conclusions? The manuscript must describe methodologically and ethically rigorous research with conclusions that are appropriately drawn based on the data presented.

Reviewer #1: Yes

Reviewer #2: Yes

Reviewer #3: Yes

2. Has the statistical analysis been performed appropriately and rigorously?

Reviewer #1: N/A

Reviewer #2: I don't know

Reviewer #3: Yes

3. Have the authors made all data underlying the findings in their manuscript fully available (please refer to the Data Availability Statement at the start of the manuscript PDF file)?

Reviewer #1: Yes

Reviewer #2: Yes

Reviewer #3: Yes

4. Is the manuscript presented in an intelligible fashion and written in standard English?

Reviewer #1: Yes

Reviewer #2: Yes

Reviewer #3: Yes

5. Review Comments to the Author

Reviewer #1: Great to see this work

1. Although the initial model is based on an already in use model which as a strength avoids cumbersome implementation and jargon https://www.ncbi.nlm.nih.gov/pmc/articles/PMC6971843/, it is still important to discuss validation further. As a model is only as good as its assumptions, it would be best to discuss caveats, potential pitfalls, and any validation or plans to validate. The modification to the seasonal forcing functions to account for the seasonal spikes could potentially have different impacts. Policy makers, though often wanting a straightforward tool without a lat of jargon, do want to see data and validation so any form of validation (on past data) or plans to validate would still be useful for policy makers.

- I would add any validation run on past data or plans for future validation or potential foreseen caveats to the discussion

- I would focus more on discussing the model itself and its strengths but also potential than the specific outputs

2. It would be helpful to discuss how other models have included seasonality. Is it possible to compare with other models that have looked at seasonality?

https://www.ncbi.nlm.nih.gov/pmc/articles/PMC9167503/

3. It would be helpful if making predictions into the 2030s to include discussion, at least, of other factors which may affect continuing progress and could discuss other

- Potential artesunate resistance (though not currently a pressing issue could reach a tipping point where becomes a factor in the next decade https://pubmed.ncbi.nlm.nih.gov/33746510/
https://www.nature.com/articles/s41598-022-11790-9

https://www.nejm.org/doi/full/10.1056/NEJMoa2101746

https://www.who.int/news-room/questions-and-answers/item/artemisinin-resistance

- ITN resistance, though often overstated, still worth mentioning https://pubmed.ncbi.nlm.nih.gov/35331237/

- Likewise IRS resistance as have needed to change in the past, though not anticipated now could have resistance more so in the coming decade(s) https://www.ncbi.nlm.nih.gov/pmc/articles/PMC7353711/

- pfhrp2/3 gene deletions (and false negative RDTs) which is a growing problem in Africa particularly the Horn of Africa, and has not proven itself much of an issue at this time in Ghana but pathogens are dynamic and mobile, so over the course of decade this could change.

https://www.who.int/news/item/28-05-2021-statement-by-the-malaria-policy-advisory-group-on-the-urgent-need-to-address-the-high-prevalence-of-pfhrp2-3-gene-deletions-in-the-horn-of-africa-and-beyond

https://www.who.int/publications/i/item/WHO-CDS-GMP-2019.02

https://journals.plos.org/plosone/article?id=10.1371/journal.pone.0238749

https://malariajournal.biomedcentral.com/articles/10.1186/s12936-019-2987-4

- Climate Change. Although the model includes El Niño and weather information, I would still make note of how climate change which may disrupt current patterns and cause greater drought, seasonal changes, even population movement away from say dry areas to wetter areas, could affect predictions in the longerterm rather than just a year or so in advance. The modification of seasonal forcing functions could behave differently if climate change acerbated.

- Mobility patterns can also affect the number in certain zones and also the immunological landscape.

Minor

Line Numbers would help a lot.

Do ensure abbreviations are defined on first use, even if commonly understood (IRS) to allow readers outside of field or sub-field to appreciate paper

Check use of it's vs its

Also 404 message on gitlab when logged in:

https://gitlab.com/tawine/SPPf_tool_Ghana_regional_analysis

Reviewer #2: This is an important article to help in planing for malaria elimination and also relevant in overall disease control planning and forecasting.

I would suggest the meaning of the acronym SMC is clarified as it is rather a chemo prevention strategy and not a chemotherapy. I also suggest that the acronyms ITN is spelt out in full at the first mention.

It may also be better to be consistent with either using insecticide treated nets or long lasting insecticide treated nets or bednets in the abstract and main article, with a clear definition of what it refers to.

I am.also of the view that the cost effectiveness of each strategy may have been useful to help in decision making. Authors did mention that a study had looked into the cost of different strategies. The discussion may have highlighted the potential cost of each modelled outcomes

Reviewer #3: The language used in the submitted article is clear, coherent and concise. The model application employed for the study is logical and convincing. The method employed tor the study is appropriate. All tables are relevant to the method and the objectives of the study. The discussion section was appropriate and sufficient. The limitation was mentioned in a justifiable way, e.g., inter-regional coverage and age distribution.

6. PLOS authors have the option to publish the peer review history of their article (what does this mean?). If published, this will include your full peer review and any attached files.

**Do you want your identity to be public for this peer review?** For information about this choice, including consent withdrawal, please see our Privacy Policy.

Reviewer #1: No

Reviewer #2: No

Reviewer #3: No

---

## [Decision Letter · Decision Letter 1]

14 Nov 2022

PGPH-D-22-00584R1

Assessing the effectiveness of malaria interventions at the regional level in Ghana using a mathematical modelling application

Dear Dr. Awine,

Thank you for submitting your manuscript to PLOS Global Public Health. After careful consideration, we feel that it has merit but does not fully meet PLOS Global Public Health’s publication criteria as it currently stands. Therefore, we invite you to submit a revised version of the manuscript that addresses the points raised during the review process.

We look forward to receiving your revised manuscript.

Kind regards,

Mathieu Nacher

Academic Editor

Journal Requirements:

Additional Editor Comments (if provided):

Reviewers' comments:

Reviewer's Responses to Questions

**Comments to the Author**

1. If the authors have adequately addressed your comments raised in a previous round of review and you feel that this manuscript is now acceptable for publication, you may indicate that here to bypass the “Comments to the Author” section, enter your conflict of interest statement in the “Confidential to Editor” section, and submit your "Accept" recommendation.

Reviewer #1: All comments have been addressed

Reviewer #3: All comments have been addressed

2. Does this manuscript meet PLOS Global Public Health’s publication criteria? Is the manuscript technically sound, and do the data support the conclusions? The manuscript must describe methodologically and ethically rigorous research with conclusions that are appropriately drawn based on the data presented.

Reviewer #1: Yes

Reviewer #3: Yes

3. Has the statistical analysis been performed appropriately and rigorously?

Reviewer #1: N/A

Reviewer #3: Yes

4. Have the authors made all data underlying the findings in their manuscript fully available (please refer to the Data Availability Statement at the start of the manuscript PDF file)?

Reviewer #1: Yes

Reviewer #3: Yes

5. Is the manuscript presented in an intelligible fashion and written in standard English?

Reviewer #1: Yes

Reviewer #3: Yes

6. Review Comments to the Author

Reviewer #1: I would consider reducing the images presented to make them more graphically readable. Others could be in the supplement.

I would consider more specifically speak to how climate change will affect malaria predictions, as such predictions will look very different in the course of a decade. I would mention the phrase climate change in the paper.

https://link.springer.com/article/10.1007/s00285-018-1229-7

From before:

Climate Change. Although the model includes El Niño and weather information, I would still make note of how

climate change which may disrupt current patterns and cause greater drought, seasonal changes, even

population movement away from say dry areas to wetter areas, could affect predictions in the longerterm rather

than just a year or so in advance. The modification of seasonal forcing functions could behave differently if

climate change acerbated.

Response: The model allows for changes to be made to the forcing function as this will be captured in

the weather data used for fitting the models. This has been included in the statement above about

changes in the environment and its impact on the predictions.

I might make mention of varying socioeconomic levels across regions, as lower socioeconomic status and household wealth, which has been correlated with higher risk of malaria infection, in discussion. Would explain more why the intervention scenarios of health systems strengthening and strengthening the CHW/CHPS have significant overlaps. Would be more specific about what health systems strengthening means from a practical standpoint and why it would be a separate intervention scenario from CHW/CHPS strengthening

https://www.ncbi.nlm.nih.gov/pmc/articles/PMC3120731/

https://www.ncbi.nlm.nih.gov/pmc/articles/PMC3483782/

From before:

Check use of it's vs its

-> Response: edited appropriately

Reviewer #3: The manuscript elucidates the importance of bed nets, indoor insecticide spraying, and diagnosis and treatment of malaria as efficient intervention tools for modeling and forecasting the pre-elimination of malaria in Ghana. Although there are other significant intervention tools for modeling and forecasting. The writer believed that those mentioned are more appropriate and applicable to Ghana. It is a novel study in Ghana, but the writer still needs more references from similar studies to prove the genuineness of his findings. I could not see clearly any limitation to this study and further opportunities to explore.

7. PLOS authors have the option to publish the peer review history of their article (what does this mean?). If published, this will include your full peer review and any attached files.

**Do you want your identity to be public for this peer review?** For information about this choice, including consent withdrawal, please see our Privacy Policy.

Reviewer #1: No

Reviewer #3: No

---

## [Editor Report · Decision Letter 2]

28 Nov 2022

Assessing the effectiveness of malaria interventions at the regional level in Ghana using a mathematical modelling application

PGPH-D-22-00584R2

Dear Dr Awine,

We are pleased to inform you that your manuscript 'Assessing the effectiveness of malaria interventions at the regional level in Ghana using a mathematical modelling application' has been provisionally accepted for publication in PLOS Global Public Health.

Best regards,

Mathieu Nacher

Academic Editor